# Multipurpose X-Ray Stage and Its Application for In Situ Poling Studies

**DOI:** 10.3390/ma18051004

**Published:** 2025-02-25

**Authors:** Antonio Iacomini, Davide Sanna, Marzia Mureddu, Laura Caggiu, Costantino Cau, Stefano Enzo, Edgar Eduardo Villalobos-Portillo, Lorena Pardo, Sebastiano Garroni

**Affiliations:** 1Electronic Ceramics Department, Jožef Stefan Institute, 1000 Ljubljana, Slovenia; antonio.iacomini@ijs.si; 2Department of Chemical, Physical, Mathematical and Natural Sciences, University of Sassari, Via Vienna 2, I-07100 Sassari, Italy; dvdsanna@uniss.it (D.S.); m.mureddu6@studenti.uniss.it (M.M.); lcaggiu@uniss.it (L.C.); c.cau1@phd.uniss.it (C.C.); enzo@uniss.it (S.E.); 3Department of Architecture, Design and Urban Planning, University of Sassari, Piazza Duomo 6, I-07041 Alghero, Italy; 4Alba Synchrotron Light Source, Carrer de la Llum 2-26, 08290 Cerdanyola del Vallès, Spain; evillalobos@cells.es; 5Instituto de Ciencia de Materiales de Madrid, CSIC, C/Sor Juana Ines de la Cruz, 3 Cantoblanco, 28049 Madrid, Spain

**Keywords:** XRD, in situ experiments, diffraction, piezoceramics, prototype, poling experiments

## Abstract

A 3D-printable, ARDUINO-based multipurpose X-ray stage of compact dimensions enabling in situ electric field and temperature-dependent measurements is put into practice and tested here. It can be routinely applied in combination with a technique of structural characterization of materials. Using high-performance X-ray laboratory equipment, two investigations were conducted to illustrate the device’s performance. The lattice characteristics and microstructure evolution of piezoelectric ceramics of barium titanate, BaTiO_3_ (BT), and barium calcium zirconate titanate, with compositions of (Ba_0.92_Ca_0.08_) (Ti_0.95_Zr_0.05_)O_3_ (BC8TZ5), were studied as a function of the applied electric field and temperature. The X-ray stage is amenable as an off-the-shelf device for a diffraction line in a synchrotron. It provides valuable information for poling piezoceramics and subsequent optimization of their performance.

## 1. Introduction

The automotive, aerospace, communications, electrical, and food industries, as well as the medical field, are just a few of the many mature fields of application in which the class of materials known as piezoelectric ceramics finds use. The emerging applications in biotechnology and energy, among others, together with a strong demand concerning their sustainability and reliability, are creating a renewed interest in the knowledge of these materials to enhance and control their performances. They are, in fact, ferroelectric polycrystals that have undergone the so-called poling process, when exposed to a high electric field, which results in an anisotropic piezoelectric response [1]. The structure of the solid is the source of the ferroelectric phenomena. The properties of ferroelectric materials are also highly dependent on the temperature and applied electric field because they have a significant effect on structure. A range of experimental methods, such as X-ray or neutron diffraction [2,3], Raman or Micro-Raman [4], and infra-red (IR) [5] spectroscopies, among others, are currently used to determine the structural evolution of ferroelectrics as a function of in situ-applied temperature or electric field. The ferroelectric domain structure can also be studied by force microscopy. It is now possible to observe domain structures, barriers, and polarization processes in real space. The study of the electromechanical coupling at the nanoscale in a range of materials, including ionics and biological systems, has been made possible by piezoelectric force microscopy (PFM), which applies the electric field in situ using conducting probes [6]. Numerous studies using the mentioned techniques have shown how important in situ measurements under temperature and applied electric field are for the study of piezoelectric ceramics.

The increasing interest in in situ X-ray diffraction (XRD) experiments using piezoceramics during the poling process is attributed to the valuable insights they offer. The poling process of the ceramic under an applied intense electric field is typically conducted above room temperature and at a temperature below the Curie temperature of the material in order to facilitate the movement of domain walls [7,8]. Investigating the effects of temperature on the polarization process yields significant advantages; it enables a thorough comprehension of the poling process, i.e., domain wall movement, and facilitates the optimization of relevant parameters, including electric field and temperature [9,10]. A detailed examination of the crystalline structure evolution in response to the temperature imposed on both polarized and non-polarized systems is also of notable significance. These explorations shed light on de-poling processes and provide insights into the crystalline structure’s temperature-dependent evolution. To further demonstrate the potentiality of these combined techniques in the field of piezoceramics, many investigations can be cited. In situ field and temperature-dependent X-ray diffraction were used [11] to investigate the impact of field cooling (FC) on the domain reorientation of a BCZT composition. In addition, other researchers discovered an intermediate monoclinic phase in textured KNN-based materials, which acts as a connecting bridge to facilitate the process of polarization rotation [12]. Non-ergodic relaxor ferroelectrics are extensively studied with this technique as their symmetry changes with the application of an external electric field [13,14]. In situ diffraction experiments are also employed in order to evaluate the percentage of the 90° domains reoriented during poling and to provide information about the relaxation effect that takes place when the poling fields are removed [15,16].

Although some of the mentioned works were performed at synchrotron facilities, experiments using a conventional laboratory diffractometer have also been widely reported in the current literature [2,7,16,17]. The experiments were performed principally to investigate the origin of the thermal de-polarization in different piezoceramics, including BCZT, and the microstructure–property relationship. However, the devices designed and used for these measurements suffer from a high cost and complex configuration, which limits their extended use. Furthermore, some of the measurement X-ray stages were developed for thin films, excluding their use for materials in bulk form [18], whereas this work has a dual target.

Starting from these considerations and in line with the increasing trend for the development of miniaturized or portable and affordable instruments based on state-of-the-art hardware manufacturing and computers [19,20,21], this work presents a multipurpose X-ray stage. A 3D-printable and ARDUINO-based device with reduced dimensions, recently patented [22], for in situ electric field and temperature-dependent measurements, is here implemented and tested. It can be routinely used in conjunction with a technique of structural determination of bulk ceramic materials. In order to demonstrate the device’s performance, two temperature- and field-dependent X-ray diffraction studies were carried out using high-performance laboratory equipment. Barium titanate, BaTiO_3_ (BT), and barium calcium zirconate titanate with a composition of (Ba_0.92_Ca_0.08_) (Ti_0.95_Zr_0.05_)O_3_ (BC8TZ5) were studied, with the results evidencing the lattice parameters and microstructure evolution as a function of the applied electric field and increasing temperatures.

## 2. Materials and Methods

### 2.1. X-Ray Stage Description

In Figure 1, representative images of the X-ray stage (hereinafter also called X-Poll, X-ray diffraction Poling Cell) and its main components are reported. The single components (Figure 1a) are here indicated as follows: (1) Two electric cables; (2) U copper shape electrode (upper electrode); the U shape was designed to facilitate electrical contact and ensure the exposure of the sample to the X-ray beam; (3) the solid body (5 L × 5 W × 3 H—cm) was 3D printed by acrylonitrile styrene acrylate (ASA); (4) copper screw (lower electrode); (5) copper screw block; and (6) stainless steel spring. The spring around the copper screw enables optimal electrical contact between the pellet and the two copper electrodes by applying a small amount of pressure The top and bottom sides of the X-ray stage are reported in Figure 1b and Figure 1c, respectively. The removable drawer in Figure 1b was designed to facilitate the allocation of the sample inside the cell. Furthermore, it protects the metal counterpart (copper screw), preventing any interaction with the X-ray diffraction beam. The cavity of the copper screw enables the accommodation of a heating cartridge as reported in Figure 1d. In our system, the acceptable geometry of the pellets is either a disc or a rectangular plate, depending on the specific characterization requirements.

Additionally, a grounding system was implemented to ensure the dissipation of any residual charge, enhancing the safety and stability of the experimental setup.

The heating system of the X-ray stage is shown in detail in Figure 2 and consists of (1) a heating cartridge (cartridge diameter: 6 mm and length: 15 mm, 5.96 Ω, 12 V, 97 W, temperature range from R.T. up to 200 °C), (2) a thermistor for temperature reference, (3) an ARDUINO (IDE 2.3.4, Turin, Italy) temperature controller circuit (12 V), and (4) the cartridge feed system (12–24 V). The ASA used for the solid body is an amorphous thermoplastic polymer often used as an alternative to a similar polymer such as acrylonitrile butadiene styrene (ABS). Compared to ABS, ASA presents better durability and good resistance including tolerance to water and UV radiation [23]. Furthermore, it possesses good mechanical properties, which are due to the excellent adhesion between layers that guarantee high impact resistance and strength during the printing process [24]. The current generator, or power supply, is provided by a Consort BVBA (EV3330 model, Turnhout, Belgium) operating with a voltage range of 300–3000 V.

Figure 3 illustrates the specific setup used for in situ temperature and poling experiments. In this work, the X-ray stage enables a sample disk to be annealed (25–140 °C) by applying an electric field (up to 30 kV/cm) while measuring the phase microstructure evolution by an X-ray laboratory diffractometer in an extended 2-Theta angle range (10–120° Cu target X-ray source). Regarding the experimental setup depicted in Figure 3, the electric field is applied along the Z-axis, where the Z-axis is considered the vertical direction (perpendicular to the copper screw). This configuration ensures the optimal alignment of the applied field with respect to the sample.

### 2.2. Material Preparation and Characterization

Synthesis details and structural and morphological characterization of the as-prepared BT and BC8ZT5 ceramics are reported in the Appendix A. Electrode deposition was made by using an automatic sputter coater (Agar Scientific, Rotherham, UK) equipped with an Au target source. A 40 nm Au layer was then deposited on the samples. Structural investigations were performed by using a SMARTLAB diffractometer (Rigaku Corporation, Tokyo, Japan) with a rotating anode source of copper (λ = 1.54178 Å) working at 40 kV and 100 mA. The diffractometer is equipped with a graphite monochromator and a scintillation tube in the diffracted beam. Additionally, it includes an automatic alignment system (Z-scan alignment), which enables minimization of the offset of the diffraction experiments. The analysis of the patterns collected, including the evolution of the microstructure parameters, was performed by the MAUD software (v. 2.9997), a Rietveld-based program [25]. Permittivity vs. temperature curves at frequencies above 1 kHz were measured using an automatic temperature-controlled electric furnace and capacitance-loss tangent data acquisition from an impedance analyzer (HP 4194A, Hewlett-Packard, Palo Alto, CA, USA). The morphology of the ceramics was characterized by scanning electron microscopy using a Quanta FEI 200 Scanning Electron Microscope (FEI Company, Hillsboro, OR, USA).

Synchrotron X-ray diffraction (XRD) analysis was carried out in transmission mode at the NOTOS beamline of the ALBA Synchrotron Light Source (Barcelona, Spain). The synchrotron light coming from a bending magnet was first vertically collimated, then monochromatized using two pairs of liquid-cooled Si(111) crystals, and finally focused on the sample position down to ~800 × 500 μm^2^. Rh stripe coating of the two mirrors was carried out to guarantee higher harmonic rejection. Energy calibration was conducted by measuring a Si pellet (SRM 640C) in the same configuration as the sample at 21 keV (0.59037 Å). The DECTRIS MYTHEN detector system (Baden-Daettwil, Switzerland) was used to collect data, averaging 5 different sample positions in order to minimize the effect of potential inhomogeneities of the ceramic sample.

Morphological characterization of the powder samples and the fresh fracture surfaces of the sintered ceramic disks was performed using scanning electron microscopy (SEM) with a Phenom Pro G2 SEM microscope (Thermo Scientific, Waltham, MA, USA), operating at a beam voltage of 5 kV.

## 3. Results

### 3.1. Barium Titanate

The diffraction patterns acquired on the BT systems thermally treated for increasing temperatures (RT—110 °C) by the manufactured cell are shown in Figure 4a. It is possible to appreciate the well-known tetragonal–cubic phase transition at around 100 °C (Figure 4b), which corresponds to the Curie point of this system.

In the current literature, the Curie temperature for barium titanate is commonly reported to be around 120 °C, and the difference observed in our system could be ascribable to the oxygen vacancy, impurities, or structural defects generated by deviations from the Ba/Ti ratio equal to 1 [26,27]. Subsequently, the experimental patterns were analyzed through the Rietveld method, and the results are reported in Figure 5 (refer to Appendix A for detailed values).

From room temperature to 90 °C, the tetragonal structure undergoes a progressive increase in the cell volume from 64.367 Å^3^ to 64.626 Å^3^. This is mainly due to the variation in a lattice parameter (a = b for tetragonal structures), which increases from 3.996 Å to 4.007 Å, while the c parameter slightly decreases from 4.031 Å to 4.025 Å. This behavior involves a progressive decrease in the c/a ratio, which means that the BT crystal structure undergoes the tetragonal–purely cubic transformation at 100 °C. In order to confirm and validate the results obtained with the heating system of the device on the BT sample, its phase transition was further investigated with dielectric permittivity measurements. The results shown in Appendix A further confirm that the phase transition for this sample occurs at around 100 °C.

A second experiment was conducted by applying the electric field on the BT sample at room temperature and at 80 °C. Figure 6 presents the results of the experiment described above, highlighting the evolution of the diagnostic *(002)* and *(200)* peaks.

As emerged from the specific literature, 180° domain reversal and 90° domain reorientation occur in perovskite ferroelectrics with tetragonal symmetry during poling [28]. In particular, the latter (90° domain) is responsible for the change in intensity of the diffraction peaks. Focusing on the diagnostic peaks reported in Figure 6a,b, it should be taken into account that, under normal conditions, BaTiO_3_ has a tetragonal structure below the Curie temperature. This structure presents a slight distortion along one of the crystallographic axes (c-axis), which leads to the formation of a permanent electric dipole. The *(002)* peak represents diffraction from planes perpendicular to the c-axis of the tetragonal cell. The *(200)* peak represents diffraction from planes perpendicular to the *a*-axes of the tetragonal cell. When an external electric field is applied to a BaTiO_3_ crystal, a phenomenon called “switching” of the ferroelectric domains occurs. Electric dipoles within the material tend to align with the applied electric field. If the electric field is applied along the c-axis, more domains will align in this direction. As a result, the intensity of the *(002)* peak increases, while that of the *(200)* peak decreases due to a greater alignment of unit cells with the c-axis parallel to the electric field, as shown in Figure 6a,b, which is consistent with findings from other studies [29].

A commonly used model takes into account the maximum intensities of the (002) and (200) peaks in order to evaluate the percentage of the 90° domains oriented with the field (Equation (1)):(1)$$N=\frac{R-r}{\left(R+1)(r+1\right)}\times 100$$
where *N* represents the percentage of reoriented 90° domains, and *R* and *r* denote the ratios of I_200_ to I_002_ in the unpoled (*R*) and poled (*r*) states, respectively. The data collected from the in situ investigation are reported in Appendix A and plotted in Figure 6c. As is clearly shown, the percentage of the 90° domains switched is in a proportional relation with the applied field, both for the experiment performed at RT and at 80 °C. Concerning the ratio I_200_/I_002_, it progressively decreases as the electric field reaches the final value of 20 kV/cm. It is worth noting that the experiment conducted at 80 °C demonstrates a higher yield in reorientation, particularly notable at 8 kV/cm, where the percentages of 90° domains switched are 4.46% at RT vs. 7.90% at 80 °C. This behavior is still observed up to 20 kV/cm, but less pronounced, where the maximum reorientation of 13.51% and 15.52% for RT and 80 °C was reached, respectively.

### 3.2. Barium Calcium Zirconium Titanate

Both materials tested in this work have important technological interests such as ferroelectric and piezoelectric ceramics. While BT was the first ferroelectric oxide studied and is still in use [30], the Ca and Zr modifications of the BT recently resulted in a high-sensitivity piezoelectric material (BC15TZ10) due to the coexistence of ferroelectric polymorphs [31]. The BC8TZ5 ceramic also shows a tetragonal structure [32], but the tetragonal distortion is much lower than the one of the pure BT. For this reason, the type of advanced X-ray studies accomplished in the previous section for BT using the X-Poll stage became a rather challenging task at a laboratory diffractometer, hence the reason for choosing this material. Also, for this reason, the starting point of this study was to measure the structure using X-ray diffraction at a synchrotron beamline (Appendix A) to strengthen the reliability of the data obtained at the laboratory diffractometer. From this measurement, it is apparent that the BC8TZ5 ceramic investigated in this study is actually a single-phase tetragonal perovskite. For the BC8TZ5 ceramic, the heating system of the X-Poll stage was used to confirm the reported transition temperature to the cubic paraelectric phase, which was determined by dielectric measurements [33,34]. In Figure 7a, the patterns acquired at room temperature (RT) and 100 °C are shown. The Rietveld refinement of the whole XRD measurements performed at RT revealed the presence of a pure *P4mm* structure with a = 4.0045 Å and a *c*/*a* = 1.004. The pattern acquired at 100 °C showed a cubic *Pm-3m* phase with a = 4.0127 Å and *c*/*a* = 1.000. As better highlighted by the diagnostic peak at 45° *(002*/*200)* of the XRD pattern collected at RT and 100 °C (Figure 7b), the crystalline cell transformation into a cubic polymorph was completed, as expected, at this temperature. Thus, the validity of the combination of the X-Poll stage and high-performance laboratory diffractometer was assessed, even in this extreme case, in which the overlapping of the tetragonal (002/200) doublet at RT is high.

Similarly, the validity of the X-Poll stage was tested to conduct measurements under an electric field application. Figure 8 shows the evolution, at room temperature, of the diagnostic doublet at 45° as a function of the poling electric field (0–12 kV/cm). The electric field applied does not affect the lattice tetragonal distortion that determines the peak’s angular position, which remains unchanged for the whole range of fields tested. The only variation is undoubtedly imputable to the decrease in intensity of the *(002)* crystalline family plane of the single tetragonal (*I4mm* s.g.) polymorph.

The increase in the background is noticeable as the electric field increases, which is probably due to the expansion of this piezoelectric high-sensitivity sample (d_33_ = 320 pC/N [32] to be compared with d_33_ = 190 pC/N for BT [30]) under the applied field.

Obviously, here the study of the percentages of the reoriented 90° domains from the data in Figure 8 requires a previous deconvolution of the diagnosis doublet, which can be easily performed nowadays with a diversity of software. This matter is outside the scope of this work. In this way, the validity of the X-Poll device is also demonstrated in studies in an applied field, even in this extreme case.

## 4. Conclusions

This work provides the “proof of concept” of a new device able to perform in situ poling/temperature diffraction experiments for piezoceramics and more. The development and engineering of the cell is described in detail. To show the potentialities of the device, two basic experiments are reported on the well-known piezoceramics of barium titanate, BT, and BC8ZT5 systems. The reliability of the results was verified with auxiliary characterizations and by comparison with the literature. We have shown that the cell is a useful tool that performs cost-effectively, while accurate, in situ diffraction experiments reveal valuable information for poling and subsequent optimization of material performance when used in conjunction with a high-performance laboratory scale diffractometer, as well as an off-the-shelf device for a diffraction line in a synchrotron. Some salient aspects, such as the wireless functionality of the temperature controller, could be subject to further refinement of the device.

## 5. Patents

The Italian Patent No. 102022000009386, titled “*Cella di polarizzazione e misura per materiali ceramici piezoelettrici*”, was filed on 6 May 2022, and granted on 23 April 2024. The patent, assigned to the Università degli Studi di Sassari, was developed by Antonio Iacomini, Davide Sanna, Pier Nicola Labate, Andrea Melis, Sebastiano Garroni, Alberto Mariani, Gabriele Mulas, and Stefano Enzo. The patent introduces an innovative polarization and measurement device for piezoelectric ceramic materials.

## Figures and Tables

**Figure 1 materials-18-01004-f001:**
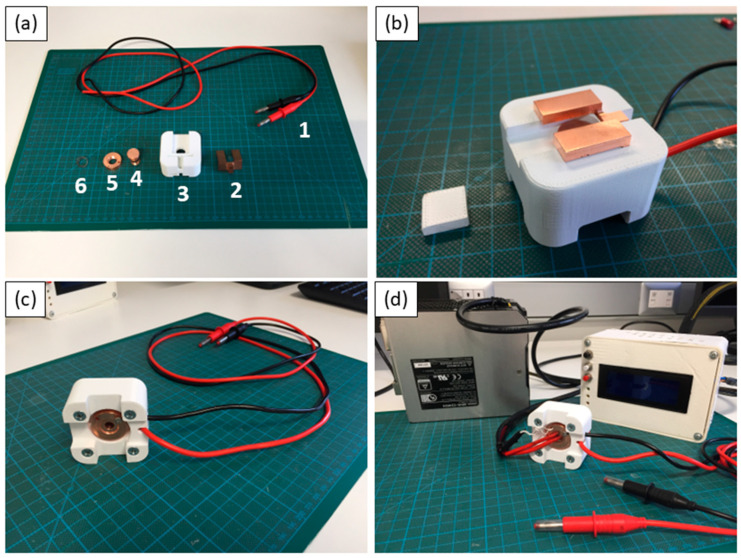
Images of the X-ray stage and its components. X-Poll before (**a**) and after (**b**) the assembly; (**c**) lower section; and (**d**) X-Poll interfaced with the heating system.

**Figure 2 materials-18-01004-f002:**
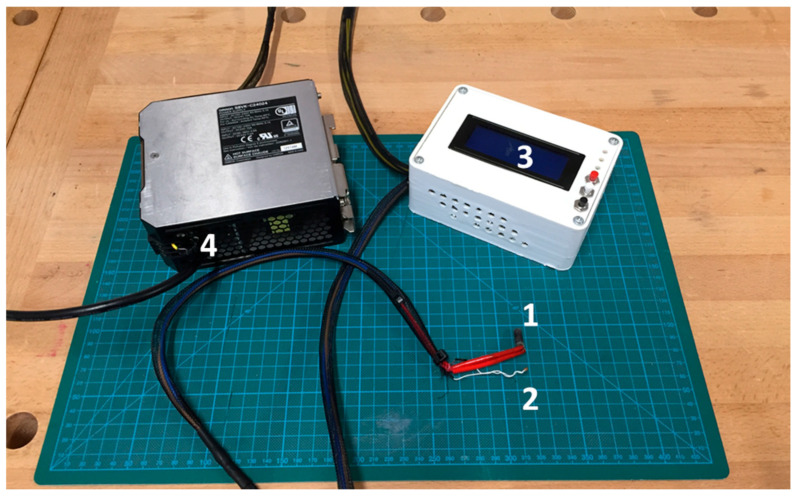
Heating system of the cell. (1) A heating cartridge (cartridge diameter: 6 mm and length: 15 mm, 5.96 Ω, 12 V, 97 W, temperature range from R.T. up to 200 °C), (2) a thermistor for temperature reference, (3) an ARDUINO temperature controller circuit (12 V), and (4) the cartridge feed system (12–24 V).

**Figure 3 materials-18-01004-f003:**
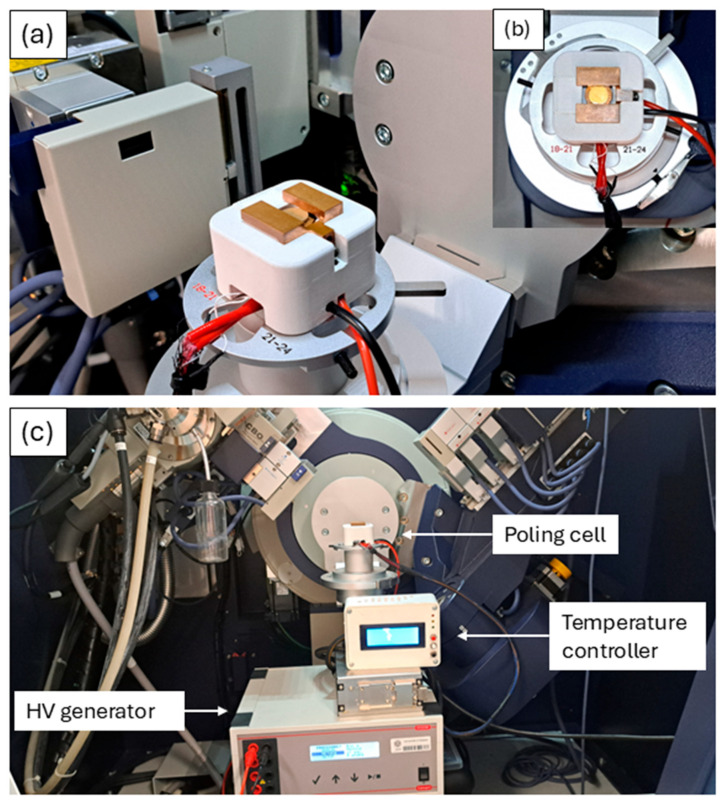
(**a**,**b**) Details of the X-Poll cell inside the diffractometer. (**c**) Image of the setup inside the diffractometer, which also includes the temperature controller and the high-voltage generator (HV generator).

**Figure 4 materials-18-01004-f004:**
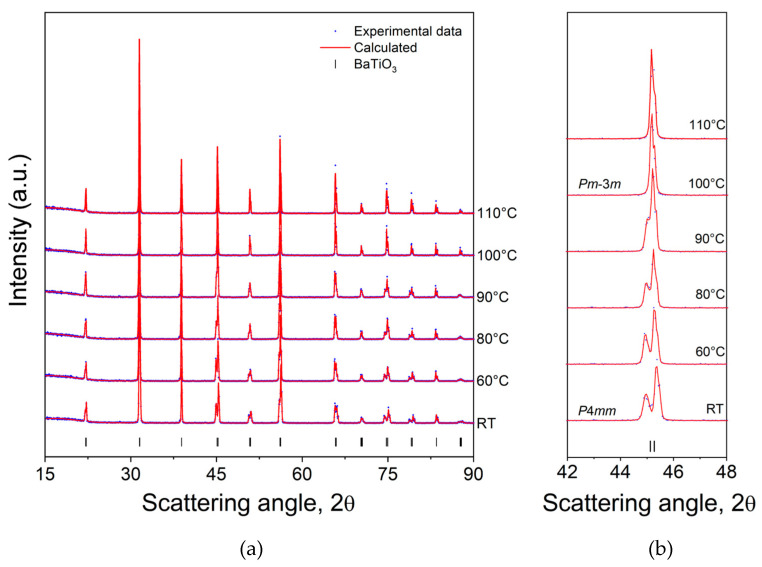
X-ray diffraction patterns and Rietveld refinement of the sintered barium titanate ceramic showing (**a**) the phase evolution as a function of temperature for the sintered BT. The blue dots are experimental data while the red line is the calculated fit. (**b**) Magnification of the diagnostic peak at around 45°.

**Figure 5 materials-18-01004-f005:**
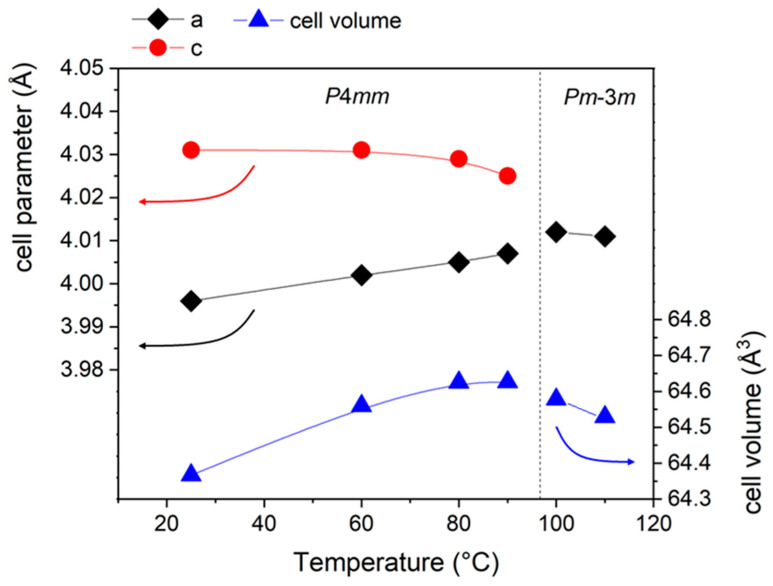
Cell parameters and cell volume evolution of the sintered BT as a function of temperature.

**Figure 6 materials-18-01004-f006:**
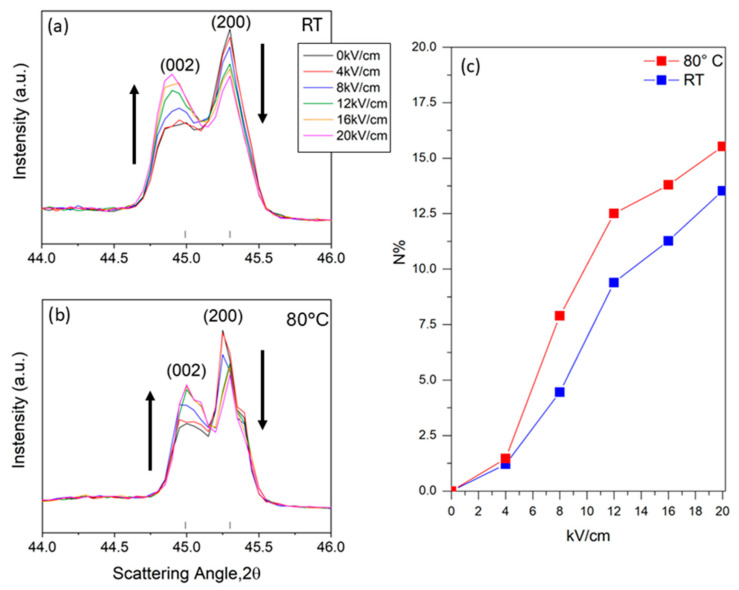
In situ electric field experiments of sintered BT. Magnification of the diagnostic peaks at 44–46°, at room temperature and 80 °C (respectively, (**a**,**b**)). (**c**) Comparison between the alignment of 90° domains at room temperature and 80 °C extrapolated from in situ experiment data. The arrows indicate the diagnostic peaks that increases and decreases as a result of the application of the electric field.

**Figure 7 materials-18-01004-f007:**
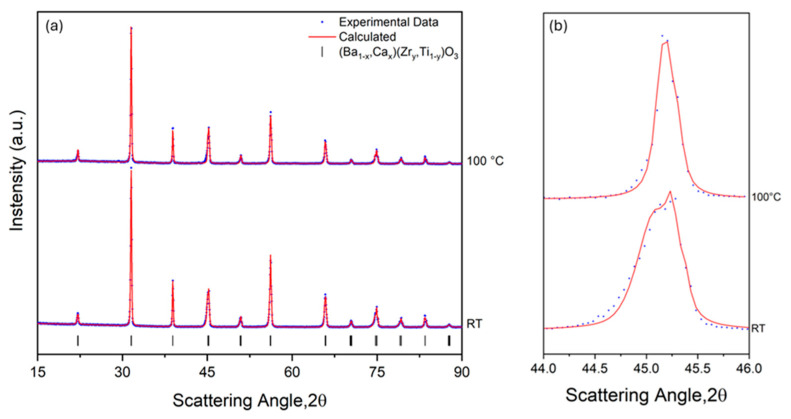
X-ray diffraction patterns and Rietveld refinement of the sintered BC8ZT5 ceramic showing (**a**) the phase evolution as a function of the temperature of the sintered ceramic at two different significant temperatures (RT and 100 °C); (**b**) the magnification of the diagnostic peak at around 45°. Data points are indicated with blue dots. The calculation from the Rietveld refinement is indicated with a red line.

**Figure 8 materials-18-01004-f008:**
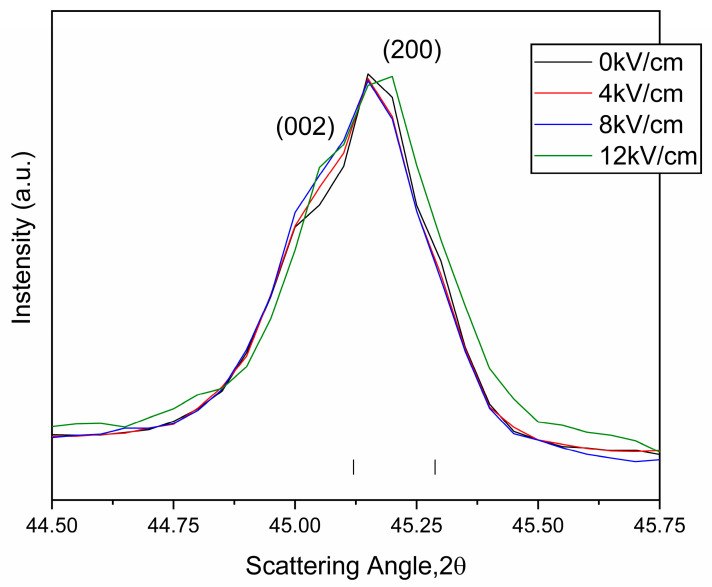
Room temperature X-ray diffraction peaks at the in situ electric field experiments of sintered BC8TZ5. Magnification of the diagnostic peak at 44–46° 2-Theta. The legend indicates the colors corresponding to the increasing electric field applied. The green-colored bar corresponds to the reflections of the BCZT *P4mm* phase in the selected angular range of 2-Theta.

## Data Availability

The original contributions presented in this study are included in the article and Appendix A. Further inquiries can be directed to the corresponding author.

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
