# Peer review of "Multipurpose X-Ray Stage and Its Application for In Situ Poling Studies"

_materials, 2025, doi:10.3390/ma18051004_

Round 1

Reviewer 1 Report

Comments and Suggestions for Authors

X-ray diffraction is particularly useful for studying structural phenomena in situ under external stimuli. Iacomini et al. describe a prototype device designed for measurement of samples while simultaneously or alternatively varying temperature and electric field. Two application cases are presented to illustrate the system’s capabilities.

The article is of quality and will likely be of interest to the piezoelectric and ferroelectric materials community. Therefore, I recommend this article for publication after addressing the following points.

Remarks

Could you specify the acceptable geometry of the pellets in our system?

Is there a system in place to discharge any residual electrostatic charges accumulated after high-voltage measurements?

Figure 3: A Bragg-Brentano configuration seems to be used in the SmartLab setup shown in the photo, i.e., measurements are performed along the qz direction. Could you clarify the direction of the applied electric field in the sample holder? The U-shaped electrical contact suggests that the field might be applied along the y-axis?

Figure 6: The 2Theta steps resolution appears coarse for individual peak fitting. In a future study, I suggest focusing on the 002/200 doublet with finer 2Theta steps and performing a two-component pseudo-Voigt deconvolution.

More generally, I suggest conducting an experiment where an electric field of 20 kV/cm is applied at a high temperature of 100°C, then progressively decreasing the temperature while maintaining the electric field. Once the field is turned off, compare if  the a/c ratio has changed.

Reviewer 2 Report

Comments and Suggestions for Authors
  1. The method of measuring sintering density of prepared BT, and BC8ZT5 ceramics should be provided in the manuscript or supplementary document.
  2. The density of the prepared BC8TZ5 pellet is only 89%, which means the pellet is not well sintered and the high porosity can affect the piezoelectric and dielectric properties.
  3. How many samples are used for each in-situ study?
  4. What is the highest temperature the Multipurpose X-ray stage can withstand?
  5. I will suggest the authors add the temperature dependent dielectric permittivity curves of the prepared BT, and BC8ZT5 ceramics to the manuscript, which can show the temperature dependent phase transition behavior and confirm the in-situ X-ray diffraction pattern results.
  6. Can this Multipurpose X-ray stage be used for thin film and powder type samples besides bulk samples?
